# Neutron Study of Multilevel Structures of Diamond Gels

**Vasily Lebedev [1],\*, Yury Kulvelis [1], Alexander Kuklin [2] and Alexander Vul [3],\***

[1]   Petersburg Nuclear Physics Institute, NRC Kurchatov Institute, 188300 Gatchina, Russia; kulvelis@pnpi.spb.ru
[2]   Joint Institute for Nuclear Research, Joliot-Curie 6, 141980 Dubna, Russia; kuklin@nf.jinr.ru
[3]   Ioffe Institute, 194021 St. Petersburg, Russia
**\***   Correspondence: vlebedev@pnpi.spb.ru (V.L.); alexandervul@mail.ioffe.ru (A.V.)

**Abstract:** The structure of a hydrogel consisting of diamond nanoparticles formed by the explosion method has been studied. Small angle neutron scattering has been used as a method for characterization of the gel. Joint approaches for data analysis in reciprocal and direct space have been developed to restore a multilevel structure. The pristine hydrogel of positively charged diamond particles (~5 nm in size, concentration ~5 wt %), even by four-fold dilution below its formation critical point, ($C^* \sim 4$ wt %) retains practically the original structure where single particles are joined into small groups integrated into chain fractal-type aggregates creating a network. This indicates a local stability of the gel and means a transformation of continuous gel into a system of micro-domains suspended in water. A perfection of the diamond crystals' facets was revealed that is of principal importance for the configuration of potentials, inducing the diamonds' electrostatic attraction due to different electric charges of facets. It is distinguished from the results for the suspensions of diamonds in graphene shells that showed a deviation of scattering from Porod's law.

**Keywords:** nanodiamond; hydrogel; structure; neutron scattering

---

## 1. Introduction

The detonation synthesis of diamonds [1–3] is associated with a problem of their removal from aggregated carbon products, because milling yields the damaged diamonds with graphene-like coatings [4,5]. Recently the method of non-destructive separation of diamonds was developed to produce desirable crystals (polyhedrons, size of 4–5 nm) [3,6,7] for modern technologies, especially for medical and pharmaceutical applications (targeted delivery of preparations, luminescent markers, isotope carriers) [8–10], which need stable aqueous systems with diamonds. Meanwhile, in neutron scattering experiments the trends of diamonds to aggregate into mass fractals at the scales of tens of nanometers were found [11–13]. In previous papers [11–13] devoted to the studies of diamonds' suspensions, the analysis was based mainly on modeling the $q$-dependencies of scattering intensities (cross sections) by some fractal laws which described the data in the corresponding intervals of momentum transfers. This approach gives a set of structural parameters resulting from the analysis in the frameworks of the chosen models. Meanwhile, such a general description cannot resolve the subtle features of particle correlations at different spatial scales. Therefore, in our work we have applied the formalism of correlation functions to restore in detail the specific features of particles' arrangements to find the characteristics of their self-assembly in hydrogels where different structural levels exist.

A transformation of suspensions into the gels is considered as a way to produce stable and controlled structuring via a connection of particles in the networks. Their stability should be provided by a fine balance of attractive and repulsive forces between the particles when they are covered by

aqueous layers preventing irreversible aggregation. Authors [7,14] have realized that the conditions of diamond hydration using a chemical treatment of diamonds' surfaces in an atmosphere of hydrogen was where the facets of crystals obtained the charges of different signs and their hydrated surfaces were protected from direct contacts. These systems show a gel formation above the critical concentration ($C^*$) which depends on the sign of the particles' potential ($C^* \sim 4$ wt % and 7 wt % for $\zeta > 0$ and $\zeta < 0$). On the other hand, for the application of gels it is necessary to save their local structure even when water content may be varied.

The present work is aimed at searching the subtle features of gels' local structure by the dilution of gel below its critical point.

## 2. Results

### 2.1. Samples and Scattering Experiments

The hydrosols of detonation nanodiamonds (DND) with positive potential ($\zeta^+$) were obtained using the following method [7]. The industrial DND-powder was annealed in hydrogen flux at 500 °C, mixed with deionized water, and sonicated. The particles of small-sized fraction were separated by centrifugation and their size distribution was controlled by dynamic light scattering (DLS). Hydrosol (concentration $C \sim 1$ wt %) in the process water evaporation in a vacuum rotary evaporator at temperature 50–60 °C was transformed into the gel with a diamond content of $C_1 = 5.05$ wt % when the critical concentration $C \sim C^* \sim 4.2$ wt % was exceeded. A detail description of the synthesis and characteristics of gels are given in previous articles [14,15]. This gel was used to obtain the systems with the lower concentrations of diamonds, $C_2 = 2.25$ wt %, $C_3 = 1.13$ wt %, by water addition (Table 1).

Neutron scattering experiments have been carried out on the Diffractometer "YuMO" (IBR-2 reactor, JINR, Dubna, Russia) in the range of momentum transfer $q = 0.06$–5 nm$^{-1}$ for the samples with diamond concentrations $C_1$, $C_2$, and $C_3$ at ambient temperature and atmospheric pressure. A thin layer of gel or diluted systems (1 mm) was used to minimize the multiple scattering effects. The data were normalized to the samples' thickness and the intensities measured for the Vanadium-standard of incoherent scattering. These data corrected for the solvent contribution gave the desirable coherent cross sections of the samples per unit solid angle and cm$^3$ of the volume.

**Table 1.** Parameters of partial correlation functions $\gamma_i(R)$ of the samples with different contents of diamond ($C$): coefficients $g_i$ and correlation radii $r_i$ ($i = 1, 2, 3$).

| $C$, wt % | 1.13 | 2.25 | 5.05 |
|---|---|---|---|
| $g_1 \cdot 10^3$, cm$^{-1} \cdot$nm$^{-3}$ | $36.4 \pm 0.1$ | $74.5 \pm 0.2$ | $154.8 \pm 0.3$ |
| $g_2 \cdot 10^3$, cm$^{-1} \cdot$nm$^{-3}$ | $14.3 \pm 0.1$ | $27.9 \pm 0.2$ | $59.0 \pm 0.5$ |
| $g_3 \cdot 10^4$, cm$^{-1} \cdot$nm$^{-3}$ | $22.8 \pm 0.4$ | $45.1 \pm 0.8$ | $94.6 \pm 1.8$ |
| $r_1$, nm | $3.51 \pm 0.01$ | $3.48 \pm 0.01$ | $3.40 \pm 0.01$ |
| $r_2$, nm | $7.12 \pm 0.03$ | $7.31 \pm 0.03$ | $7.19 \pm 0.03$ |
| $r_3$, nm | $19.7 \pm 0.1$ | $20.2 \pm 0.1$ | $19.9 \pm 0.1$ |

### 2.2. Discussion

#### 2.2.1. Dependencies of Cross Sections on Momentum Transfer

The analysis of neutron data combines the structural models' application in reciprocal space and rebuilding of the correlation functions to determine more exactly the structures of the original gel and its derivatives. As seen in Figure 1, the scattering cross sections $\sigma(q)$ for pristine gel and diluted systems grow by four orders in magnitude at low momentum transfers, that indicates an assembly of particles at the scales of $R \sim \pi/q\varepsilon \geq 10^1$ nm. At the same time, the dilution leads to the decrease of cross sections $\sigma(q)$ according to the decrement of concentration, while at first glance, the behavior of

$\sigma(q)$ does not change. Hence, the gel's ordering achieved above the critical point ($C_1 > C^* \sim 4.2$ wt %) is rather stable and exists after two- and four-fold dilution.

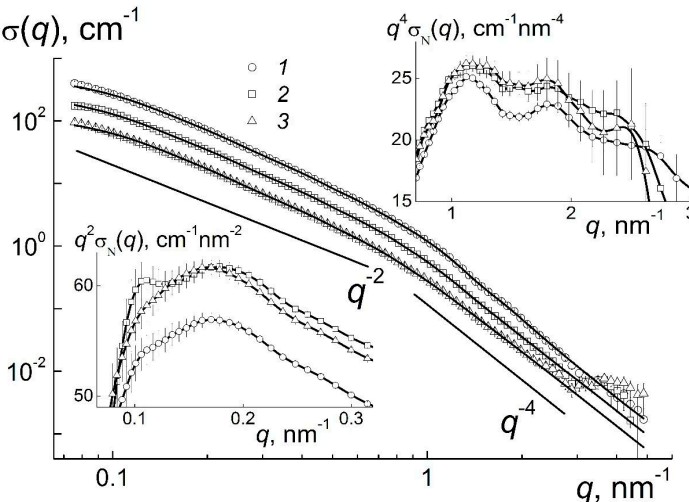

**Figure 1.** Cross sections $\sigma(q)$ for the samples with diamond contents of 5.05, 2.25, and 1.13 wt % (1–3) vs. momentum transfer. Straight lines show characteristic slopes for cross sections' $q$-dependencies. The fitting curves for the model of spherical particles are plotted also. Inserts show the Kratky- and Porod-plots (left, right) for the cross sections $\sigma_N(q)$ normalized to the diamonds' concentrations.

The retention of stable local structures below the critical point $C^*$ can be treated as a disintegration of the original gel network into large domains in water. Their size exceeds the value $\sim 2\pi/q_{\min} \sim 10^2$ nm that is defined by the minimal momentum transfer $q_{\min}$. Such a micro-gel exists in the range of concentrations from $C^*/4 \leq C \leq C^*$. It is remarkable that all the curves of $\sigma(q)$ demonstrate a kink at the momentum transfer $q \sim q^* \sim 1$ nm$^{-1}$ (Figure 1). Obviously, at $q \geq q^*$ the scattering on single particles dominate, whereas at $q \leq q^*$ the interference is observed in scattering at the distances $2\pi/q \geq d_s$ exceeding the particles' diameter $d_s \sim 2\pi/q^* \sim 6$ nm. In these regions the cross sections show the exponential behaviors, $\sigma(q) \sim 1/q^D$, with the parameters $D = D_1 \sim 4$ and $D = D_2 \sim 2$ where the first one corresponds to Porod's law observed for the particles with sharp borders and the second one indicates chain-like aggregates of particles (Figure 1).

The existence of low-dimensional structures, i.e., linear (chain-like) aggregates is clearly visible from the q-dependence of the cross section, $\sigma(q) \sim 1/q^2$, since this scattering law is established from the statistics of flexible chains (polymers) [16]. At the same time, except for linear fragments, the observed structures may include some branched (e.g., dendrimer-like) fragments characterized by the exponent $D_2 > 2$ [16,17].

The logarithmic presentation (Figure 1) has revealed the basic motives of gel structuring. The subtle features of the systems are visible by a compensation of the exponential dependencies when we used the Kratky- and Porod-plots at $q \geq q^* \sim 1$ nm$^{-1}$ and $q < q^*$ for the cross sections $\sigma_N(q) = \sigma(q)/C$ normalized onto the diamonds' concentrations, given in Figure 1 (inserts).

The right inset in Figure 1 illustrates the spacing ($d_{int}$) between the centers of neighboring particles coupled in network structure of gel. The spacing $d_{int}$ defines the position of the peak, $q_{int} \approx 1.1$ nm$^{-1} = X/d_{int}$ where the coefficient $X$ corresponds to the maximum $X = q_{int}d_{int} \approx 7.72$ of Debye scattering functions $\frac{\sin(qd_{int})}{qd_{int}}$ for a pair of particles at the distance $d_{int}$. The value of $d_{int} \approx 7$ nm exceeds a characteristic diameter of a diamond particle, $d_{int} - d_S \approx 1$ nm, i.e., the particles in the network structure of gel contact via water shells with a thickness of two molecular layers. The first ($q_{int}$) and second maxima ($\sim 2q_{int}$) on the curves for different concentrations confirm a short range order of particles in gels.

On the other hand, the data at low momentum transfers (Figure 1, left insert) testify that the structuring at higher distance $D_w = X/q_w \approx 40$ nm where $q_w \approx 0.18$ nm$^{-1}$ is the maximum position for

a broad peak at $q \sim 0.1$–$0.3$ nm$^{-1}$. This spacing should be attributed already to the size of the cells in gels' multilevel structures, which are discussed below.

This analysis was extended to obtain more detailed information on the quality of the particles' surface and the features of their association. The approximations of the data (Figure 1) with the exponential law

$$\sigma(q) = J_i/q^{Di}, \quad i = 1, 2 \tag{1}$$

have given the coefficients $J_i$ showing the scattering ability of the observed objects and the parameters $D_i$ which characterize their geometry (Figure 2).

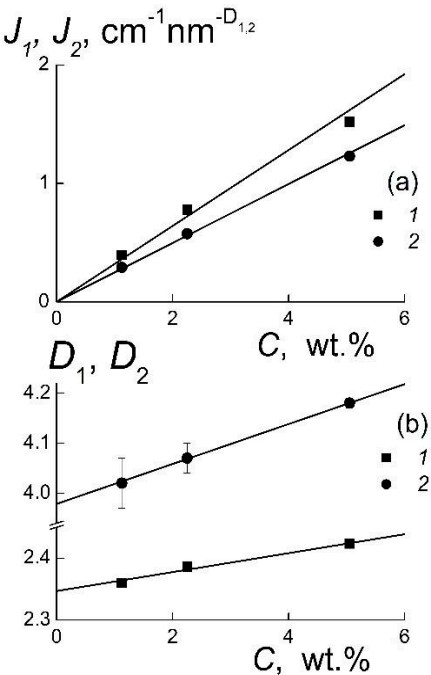

**Figure 2.** Parameters of function (1): (**a**) $J_1$, $J_2$—the characteristics of the scattering ability of fractal structures and single particles; (**b**) $D_1$—the fractal dimension of aggregates, $D_2$—the exponent describing the geometry of particles' surface.

A linear behavior of the coefficients $J_1$, $J_2$ indicates a change of the amount of particles' or aggregates by the variation of concentration, when their geometry does not undergo any substantial transformation (Figure 2a). Only a weak linear decrease of the geometric parameters $D_1$, $D_2$ is observed (Figure 2b). The $D_2$ reflects the geometry of particles' surface. This parameter declines by dilution, and in the limit, $C \rightarrow 0$, the value of $D_2 = D_{02} = 3.98 \pm 0.04$ is very close to the exponent in Porod's law describing the scattering for the particles with sharp borders [18]. It confirms a perfection of the diamond crystals' facets that is of principal importance for the configuration of potentials inducing the diamonds' electrostatic attraction due to different electric charges of the facets [19,20]. Meanwhile, even at a moderate concentration ($C_3 = 5.05$ wt %) the scattering from the particles' surface is disturbed due to the interference in scattering from neighboring particles that gives $D_{02} > 4$ (Figure 2b). This must be distinguished from the results [5,12] for the suspensions of diamonds in graphene shells, which showed a deviation of scattering from Porod's law.

The sharp surfaces of diamonds facilitate their association into branched structures evidenced from the values of $D_1 \sim 2.3$–$2.4 > 2$ (Figure 2b). Such a magnitude in the range $2 < D_1 < 3$ is inherent in the structures formed via a diffusion limited aggregation (DLA) which creates mass fractals [17]. For pristine gel the fractal dimension $D_1 \sim 2.4$ exceeds the parameter for a linear Gaussian polymer chain $D_G = 2$ [16], that confirms the existence of branched aggregates forming a network of gel. A dilution of gel retains mostly the exponent $D_1$ decreasing linearly with concentration (Figure 2b).

In the limit $C \to 0$, the extrapolated $D_1 = D_{01} = 2.35 \pm 0.01$ is substantially greater than the $D_G = 2$ for linear chains. This shows a good stability of local structure of gel grown at the volume fraction of diamonds $\phi_1 = 1.44\%$ keeping in its integrity at their low content, $\phi_3 = 0.22\%$. It is noticeable that this ordering is realized in the ensembles of particles with relatively broad size-distributions. For a comprehensive analysis of these structures, it is necessary to rebuild the size-distributions of diamonds.

### 2.2.2. Size-Distributions of Diamonds

To get the size distributions of the diamonds, the data (Figure 1) were approximated by the sum of the scattering function for spheres [18]. Therefore, the volume fractions $\Phi(R)$ of objects in the range of radii $R$ = 0–30 nm were evaluated (Figure 3). The spectra $\Phi(R)$ demonstrate a broad peak with the maximum at $R = R_{1m} \sim 2.4$–2.6 nm that agrees well with the DLS data for the diamonds prepared by the described method [14].

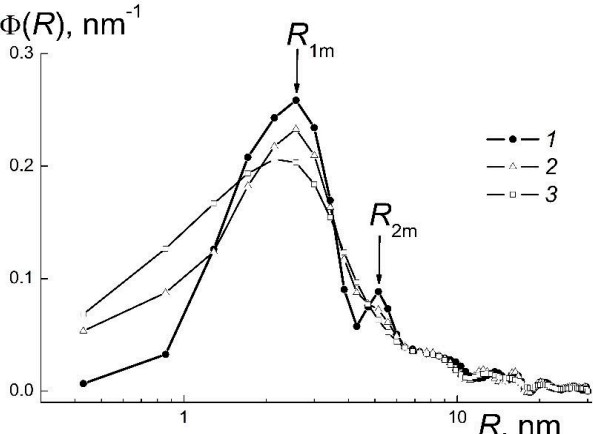

**Figure 3.** Volume fractions $\Phi(R)$ of scattering objects vs. their radii: diamonds' content of 5.05, 2.25, and 1.13 wt % (1–3). Arrows indicate the main ($R_{1m}$) and secondary maxima ($R_{2m}$).

Except for the main peak for single particles ($R_{1m} \sim 2.6$ nm), in the spectrum of the gel (Figure 3) the additional peak is seen at $R_{2m} \sim 5.1$ nm $\sim 2R_{1m}$. Its position lies in the range of radii comparable to the particles' diameter. This indicates the particles' association within the first coordination sphere. The dilution of the gel leads to disappearance of the secondary peak which overlaps with the first one that makes it complicated to distinguish single particles and small aggregates. Besides, all the samples show an aggregation at distances of double or triple the diameter of particle, $R \sim 10$–15 nm (Figure 3). To describe the ordering of the particles in gel, this model may serve as the first approximation which considers only spherical objects which scatter independently. However, various spatial correlations of particles and their aggregates should be taken into account.

### 2.2.3. Correlation Functions

The ensembles of diamond particles are characterized by the functions $\gamma(R)$ showing the correlations between various scattering centers (atomic nuclei, particles, aggregates). The function $\gamma(\boldsymbol{R}) = <\delta\rho(\boldsymbol{r})\cdot\delta\rho(\boldsymbol{r} + \boldsymbol{R})>$ is the averaged product of the deviations, $\delta\rho(\boldsymbol{r}) = \rho(\boldsymbol{r}) - <\rho>$, $\delta\rho(\boldsymbol{r} + \boldsymbol{R}) = \rho(\boldsymbol{r} + \boldsymbol{R}) - <\rho>$, in scattering length densities from the mean value $<\rho>$ in the sample. A pair correlation is considered for two points at the distance $R$. The magnitude of $<\rho> = \Sigma(b_i N_i)$ is defined by the contributions of nuclei with scattering lengths $b_i$ and concentrations $N_i$. For isotropic samples, the functions $\gamma(R)$ being the Fourier-transforms of scattering data [18] depend on the modulus of vector $\boldsymbol{R}$,

$$\gamma(R) = (1/2\pi)^3 \int \sigma(q)[\sin(qR)/(qR)]4\pi q^2 dq. \tag{2}$$

The $\gamma(R)$ for the samples are similar and demonstrate three regions of behavior (Figure 4).

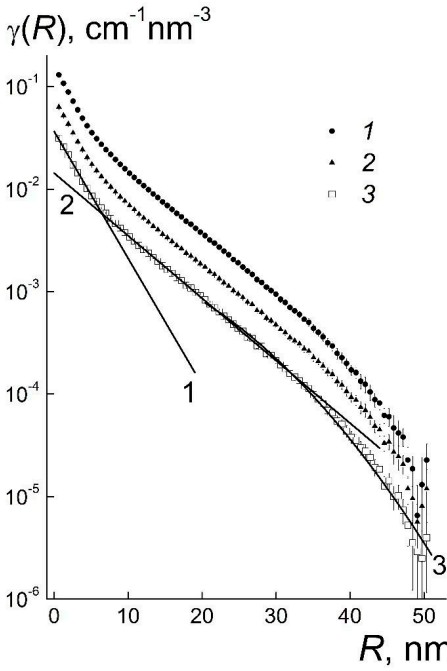

**Figure 4.** Functions $\gamma(R)$ for the samples with 5.05, 2.25 and 1.13 wt % of diamonds (1–3). The lines show the fitting functions $\gamma_1, \gamma_2, \gamma_3$ (1–3) for the lowest concentration (see text).

When the radii, $0 \leq R \leq 6$ nm, are less than the diameter of the particles, the functions $\gamma(R)$ obey the exponential law $\gamma_1(R) = g_1\exp(-R/r_1)$. It includes the correlation length $r_1$ comparable to the radius of particles and the coefficient $g_1$ proportional to the concentration of diamonds (Table 1). Since the functions $\gamma_1(R)$ describe the atomic correlations inside the particles, the parameter $r_1$ is the averaged correlation radius of particles $r_1 \sim R_{1m}$ which is slightly larger than their geometric radius (Figure 3). This follows from the comparison of $\gamma_1(R)$ and the correlation function $\gamma_S \sim [1 - (3/4)(R/r_S) + (1/16)(R/r_S)^3]$ of a sphere [18] with the radius $r_S = R_{1m}$.

At low radii, $R \leq r_1$, the $\gamma_1(R)$ is approximately a linear function, $\gamma_1 \sim [1 - R/r_1]$. At the same time, the linear term $\gamma_S \sim [1 - (3/4)(R/r_S)]$ dominates in the correlation function $\gamma_S(R)$ at $R \leq r_S$. The identity $\gamma_1(R) \equiv \gamma_S(R)$ is fulfilled if $r_S = (3/4)r_1$. Thus, in gel the particles with a correlation radius $r_1 = 3.40 \pm 0.01$ nm have the geometric radius $r_S = (3/4)r_1 \approx 2.6$ nm $\approx R_{1m}$ in accordance with experimental data (Figure 3). In addition, the $\gamma_1(R)$ yields the information on the contrast factor of diamonds $\Delta\rho_D$ relative to the surrounding medium. The $\Delta\rho_D$ depends on the scattering lengths and the concentrations of chemical elements in the particles and surroundings. This gives the indication of how tightly packed the particles are. In pristine gel with the volume fraction of diamonds $\phi = 1.44\%$, the parameter $g_1 = (\Delta\rho_D)^2\phi$ (Table 1) affords the estimate $\Delta\rho_D = (10.36 \pm 0.01) \times 10^{10}$ cm$^{-2}$. The $\Delta\rho_D = \rho_D - <\rho> = (1 - \phi)(\rho_D - \rho_W)$ is the difference between the scattering length densities $\rho_D$ of the carbon material and a similar parameter $<\rho>$ for the whole volume of the sample $<\rho> = \phi\rho_D + (1 - \phi)\rho_W$, where $\rho_W = -0.56 \times 10^{10}$ cm$^{-1}$ is the scattering length density for light water.

The obtained scattering length density of particles $\rho_D = \Delta\rho_D/(1 - \phi) + \rho_W = (9.95 \pm 0.01) \times 10^{10}$ cm$^{-2}$ is 15% lower than a similar parameter, $\rho_{DI} = 11.7 \times 10^{10}$ cm$^{-2}$, for the crystals of diamonds with a characteristic density of 3.5 g/cm$^3$. The deficit can be explained by the presence of volume (surface) defects (vacancies) in diamonds. On the other hand, we neglected the possible influence of the gel's structure factor, taken as $S(q \to 0) = 1$. Even at a low volume content of particles, their coordination may cause a deviation of this factor from unity, $S(q) < 1$.

In these systems the structuring is revealed at the radii, $6 \leq R \leq 20$ nm, comparable to the particles' diameter (first coordination sphere) that is described by the exponential function, $\gamma_2(R) = g_2\exp(-R/r_2)$. The coefficient $g_2$ is proportional to the particles' concentration but the

correlation radius $r_2$ ~ 7 nm remains almost constant (Table 1). This confirms a stability of the local ordering even by a substantial dilution of gel.

For the functions $\gamma_1(R)$, $\gamma_2(R)$ describing the correlations inside the particles and in the first coordination sphere, one can compare the correlation volumes, $V_1 = 4\pi \int \exp(-R/r_1)R^2 dR = 8\pi r_1^3 \approx$ 990 nm$^3$ and $V_2 = 8\pi r_2^3 \approx 9.3 \times 10^3$ nm$^3$. The volume $V_2$ exceeds $V_1$ by an order of magnitude but the first coordination sphere is not completely filled with particles. This is evident from a comparison of the forward cross sections $\sigma_2$ and $\sigma_1$ calculated in the limit $q \to 0$ using the functions $\gamma_1$, $\gamma_2$.

The cross section $\sigma_1 = (\Delta\rho_D)^2 \phi V_1 = g_1 V_1$ is defined by the parameters of single particles while the magnitude of $\sigma_2 = (\Delta\rho_D)^2 \phi(mV_1) = g_2 V_2$ is proportional to the number of particles ($m$) within the correlation volume $V_2$. The ratio of cross sections gives the value $m = (g_2 V_C)/(g_1 V_1) = 3.6 \pm 0.1$. Hence, a particle in the gel is connected with two to three neighboring particles. This short linear or branched fragment (aggregation number m) is really stable even in the case of a four-fold dilution of gel where only a slight reduction in the aggregation number is observed (~10%).

The structure of gel outside the first coordination sphere is seen in the $\gamma(R)$ behaviors which obey the law $\gamma_3(R) = g_3 \exp[-(R/r_3)^2]$ at $R \geq 20$ nm. It corresponds to Guinier's function $\exp[-(qR_G)^2/3]$ with the gyration radius $R_G = (3/4)^{1/2}r_3$ being proportional to the correlation length $r_3$. In gel and diluted systems, the lengths $r_3$ ~ 20 nm are the same (Table 1), as well as the gyration radii $R_G = (3/4)^{1/2}r_3$ ~ 17 nm. One may imagine the structure of these samples being composed of spherical domains having the diameter of ~$2R_G$ ~ 40 nm.

This approximation is appropriated at $q \leq 0.1$ nm$^{-1}$ (Figure 1) while in the extended interval, $q \leq 1$ nm$^{-1}$, these domains are characterized as chain fractal aggregates (Table 1). The distribution of distances between the particles inside them is described by the function $G(R) = \gamma_3(R)R^2$ having the maximum at $R = r_3$ as the most probable distance between the units in aggregates. For a Gaussian chain connecting two particles at the distance $r_3$ ~ 17 nm, the number of units is equal to $n = (r_3^2/d_S^2) + 1 \approx 16$, and the contour length $L_C = nd_S \approx 80$ nm. The latter should be compared to the spacing between the chains in the gel that is set by their numerical concentration, $N_L = \phi/(n\pi d_S^3/6) \approx 1.2 \times 10^{16}$ cm$^{-3}$ for the particles' volume fraction, $\phi = 1.44\%$. For uniform distribution of the chains in the gel, the average distance between them, $D_w \approx N_L^{-1/3} \approx 40$ nm ~ $2r_3$, is approximately half of the contour length. The chains principally overlap and form the interconnected structure of the gel with cells' diameter ~$D_w$ ~ 40 nm. This is illustrated in Figure 5 where three levels of gel structure are shown. Single particles (size $r_1$ ~ 3 nm, the first level) create small linear (branched) fragments (correlation radii $r_2$ ~ 7 nm, the second level) associated into chain structures (size $r_3$ ~ 20 nm) forming the cells (diameter $D_w$ ~ $2r_3$ ~ 40 nm) which are integrated into a gel network.

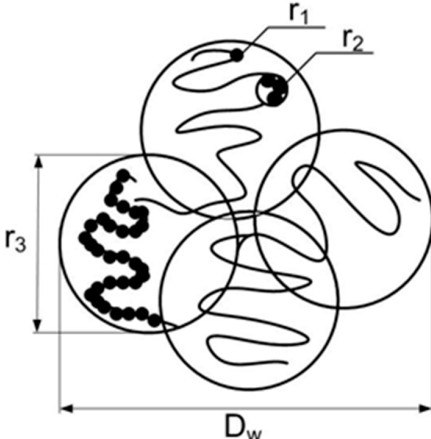

**Figure 5.** Structural levels in gel: the single particles (correlation radius $r_1$), the first coordination sphere (radius $r_2$), and the chains (size $r_3$) associated into the cells (diameter $D_w$).

In the network, the junctions' functionality $f_C = m_L/n$ can be found using the amount of particles ($m_L$) per a cell. This value is included in the forward cross section $\sigma_3 = (\Delta\rho_D)^2 \phi(\pi d_S^3/6)m_L S_L = g_3 V_L$

which is defined by the parameters of the correlation function ($g_3$, $r_3$) where $V_L = 4\pi \int \gamma_3(R)R^2 dR = \pi^{3/2}r_3^3$ is the correlation volume and $S_L(q \to 0)$ is the structure factor of the systems of the cells. The model of random contacts of spherical objects in the sample volume [18] was used to evaluate the $S_L = 1 - 8V_L/V_{1L}$, where $V_{1L} = f_C/N_L$ is the volume per cell that is determined by the numerical concentration of the cells ($N_L$). The parameter $V_L = f_C V_n/\beta$ includes the volume of the chain's region, $V_n = \pi r_3^3/6$, and the coefficient β characterizing the density of the chains' package in a cell. If the regions of chains fill the total volume, β = 1. If between the regions any cavities exist, β < 1, e.g., in the case of spheres, the most dense packing $\beta = \beta_S \approx 0.74$. Further we used the average $\beta_a = (1 + \beta_S)/2 \approx 0.87$ to estimate the $S_L = 1 - 8(\phi/n\beta_a)(r_3/d_S)^3 \approx 0.54$. Under these assumptions the amount of particles in a cell and the functionality of the network junctions is determined, $m_L = (g_3 V_L)/[(\Delta\rho_D)^2\phi(\pi d_S^3/6)S_L] \approx 70$ and $f_C = m_L/n \approx 4$.These parameters are attributed to the original and diluted systems, saving almost their initial structural features at different scales (from one to tens of diameter of particle) characterizing the multilevel order detected in gels.

## 3. Conclusions

Combined neutron scattering treatment in direct and reciprocal space has allowed us to discover the subtle structural features of diamonds' hydrogels at three levels. The formation of short range order in the ensembles of diamonds was established, primarily the aggregation of particles within the first coordination sphere around a particle and secondarily the association of tiny fragments into extended chain aggregates creating a network of gel. At the macroscopic scale, the formation of gel was detected at the critical concentration when a giant increase of suspension viscosity was detected.

It would be useful to mention that increasing the viscosity of hydrosol of the nanodiamond particles was already previously detected [4]. However that experiments used nanodiamond particles obtained by the milling method [6] having a $sp^2$ phase on the surface of the diamond particles. In the case of hydrosol produced by another method [14], we have observed a giant increase of viscosity at a concentration of diamond particles more than 4–5 wt % [21].

In neutron experiments, it was found that a dilution of gel below the critical point does not destroy the local structure of the gel. This phenomenon can be treated as a disintegration of a continuous gel into microscopic domains, which save their intrinsic structure despite the water surrounding. These results confirm the concept of the main role of electrostatic attraction of hydrated diamonds with different charges on their facets, predicted by [19,20] in the formation of gel consisting of nanodiamond particles.

**Acknowledgments:** The work is supported by the Russian Foundation for Basic Research (grant No. 14-23-01015 ofi-m) in part for neutron scattering studies. Alexander Vul thanks the support of the Russian Science Foundation (project 14-13-00795) in part for preparation and research of the properties of nanodiamond gel.

**Author Contributions:** Vasily Lebedev and Alexander Vul promoted the main idea of the manuscript, wrote the main manuscript text, and prepared all figures. Yury Kulvelis and Alexander Kuklin provided measurements, analysis, and discussion of small angle scattering data. All authors discussed the results and reviewed the manuscript. All authors have read and approved the final manuscript.

**Conflicts of Interest:** The authors declare no conflict of interest.

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
