# Peer review of "Neutron Study of Multilevel Structures of Diamond Gels"

_condensedmatter, doi:10.3390/condmat1010010_

Reviewer 1 Report

The paper of V. T. Lebedev, et al. utilizes the method of small angle neutron scattering to investigate the multilevel structures of  hydrogel at three different concentrations (with one above the critical). By analyzing the scattering spectrum, the authors obtained the size distributions of nanodiamond particles and got the conclusion that the dilution of gel below the critical point does not destroy the local structure of gel but only disintegrates the continuous gel into the microscopic domains. The authors make a very deep and comprehensive analysis on the results of neutron scattering experiment. In my opinion, the paper could be published after minor revisions. Detailed comments are listed below.

1. It is necessary to make the denotations especially some subscripts  more clearly to facilitate the reading of this paper.

2. Introduction: In my point of view, it is better to give more detailed information about other related papers and explain the uniqueness of this work.

3. Figure 1: Have you obtained the scattering cross section via momentum for the pure water? It is more rigorous  for the analysis of the particle scales to add the water scattering curve as the background.

4. Line 86: Is it possible to be dendrimer-like aggregate or other situations? Please explain more about the certainty of "Chain-like".

5. Line 105: Does y(q) denote σ(q)? Please make it clear.

6. Line 254: Is there any evidence in the second part or other references to support your statement of "a giant increase of suspension viscosity was detected"?

Author Responses

Answers for Comments 1 and 2

Thanks for the advice and suggestion,  the following addition was made in Introduction

In previous papers [11-13] devoted to the studies of diamonds’ suspensions the analysis was based mainly on the modeling the q-dependencies of scattering intensities (cross sections) by some fractal laws which described the data in the corresponding intervals of momentum transfers. This approach gives a set of structural parameters resulting from the analysis in frames of the chosen models. Meanwhile, such a general description cannot resolve the subtle features of particles correlations at different spatial scales. Therefore, in our work we have applied the formalism of correlation functions to restore in detail the specific features of particles’ arrangements to find the characteristics of their self-assembly in hydrogels where different structural levels exist.

Answers for Comment 3

Indeed, the scattering from pure water (H20) was measured and then its contribution was subtracted from the data for the samples. For data treatment in reciprocal space we also use the fitting of the total cross section σ(q) + Bg which includes the coherent part σ(q) and incoherent background of solvent, Bg. In this paper we use another way, i.e. the application of ATSAS package to obtain the distribution of the particles over the radii and to restore the correlation functions. For these purposes only the coherent part s(q) is needed.

Answers for Comment 4

Thanks for the suggestion the following addition has been made:

The existence of low-dimensional structures, i.e. linear (chain-like) aggregates is clearly visible from the q-dependence of cross section, σ(q) ~1/q2, since this scattering law for them is established from the statistics of flexible chains (polymers) [deGennes P-G. Scaling concepts in polymer physics. Cornell University Press: Ithaca and London, 1979]. At the same time, except for linear fragments, the observed structures may include some branched (e.g. dendrimer-like) fragments characterized by the exponent D2 > 2.   

Answers for Comment 5

Thanks for the suggestion it was misprint and we made correction

The misprint is corrected in eq.(1):  σ(q) = Ji/qDi , i=1,2

Answers for Comment 6

Thanks for suggestion.

Increasing of viscosity of hydrosol of nanodiamond particles has already been detected  and corresponding reference [4] is cited in the manuscript. However, experiments used nanodiamond particles obtained by milling method [6] having sp2 phase on surface of diamond particles. In the case of hydrosol produced by another method [14] we have observed a giant increase of viscosity at concentration of diamond particles more than 4-5 wt.%.

An addition and reference were made in conclusion.

Reviewer 2 Report

Interesting study on diamond gels with some high quality data. Although some of these have been described before, for the benefit of the reader, the only change I recommend is that the authors give more details on the hydrosol system. What is it, how is the gel prepared.

Author Response

Answers. Thanks for the high estimation of the manuscript. An information about the hydrogel preparation was added in paragraph 2.1, however we cited in manuscript papers for detailed information[14,15] .

Gel was prepared by evaporation of hydrosol in a vacuum rotary evaporator at temperature 50-60 0 C. This temperature does not affect the stability of the hydrosol.

Round  2

Reviewer 1 Report

The authors have made the revisions as required. In my opinion, this paper can be published.